# Understanding Local Minima in Neural Networks by Loss Surface Decomposition

## Abstract

To provide principled ways of designing proper Deep Neural Network (DNN) models, it is essential to understand the loss surface of DNNs under realistic assumptions. We introduce interesting aspects for understanding the local minima and overall structure of the loss surface. The parameter domain of the loss surface can be decomposed into regions in which activation values (zero or one for rectified linear units) are consistent. We found that, in each region, the loss surface have properties similar to that of linear neural networks where every local minimum is a global minimum. This means that every differentiable local minimum is the global minimum of the corresponding region. We prove that for a neural network with one hidden layer using rectified linear units under realistic assumptions. There are poor regions that lead to poor local minima, and we explain why such regions exist even in the overparameterized DNNs.

## 1 Introduction

Deep Neural Networks (DNNs) have achieved state-of-the-art performances in computer vision, natural language processing, and other areas of machine learning (LeCun et al., 2015). One of the most promising features of DNNs is its significant expressive power. The expressiveness of DNNs even surpass shallow networks as a network with few layers need exponential number of nodes to have similar expressive power (Telgarsky, 2016). The DNNs are getting even deeper after the vanishing gradient problem has been solved by using rectified linear units (ReLUs) (Nair & Hinton, 2010). Nowadays, RELU has become the most popular activation function for hidden layers. Leveraging this kind of activation functions, depth of DNNs has increased to more than 100 layers (He et al., 2016).

Another problem of training DNNs is that parameters can encounter pathological curvatures of the loss surfaces prolonging training time. Some of the pathological curvatures such as narrow valleys would cause unnecessary vibrations. To avoid these obstacles, various optimization methods were introduced (Tieleman & Hinton, 2012; Kingma & Ba, 2015). These methods utilize the first and second order moments of the gradients to preserve the historical trends. The gradient descent methods also have a problem of getting stuck in a poor local minimum. The poor local minima do exist (Swirszcz et al., 2016) in DNNs, but recent works showed that errors at the local minima are as low as that of global minima with high probability (Dauphin et al., 2014; Choromanska et al., 2015; Kawaguchi, 2016; Safran & Shamir, 2016; Soudry & Hoffer, 2017).

In case of linear DNNs in which activation function does not exist, every local minimum is a global minimum and other critical points are saddle points (Kawaguchi, 2016). Although these beneficial properties do not hold in general DNNs, we conjecture that it holds in each region of parameters where the activation values for each data point are the same as shown in Figure 1. We prove this for a simple network. The activation values of a node can be different between data points as shown in Figure 1, so it is hard to apply proof techniques used for linear DNNs. The whole parameter space is a disjoint union of these regions, so we call it loss surface decomposition.

Using the concepts of loss surface decomposition, we explain why poor local minima do exist even in large networks. There are poor local minima where gradient flow disappears when using the ReLU (Swirszcz et al., 2016). We introduce another kind of poor local minima where the loss is same as that of linear regression. To be more general, we prove that for each local minimum in a network, there exists a local minimum of the same loss in the larger network that is constructed by adding a node to that network.

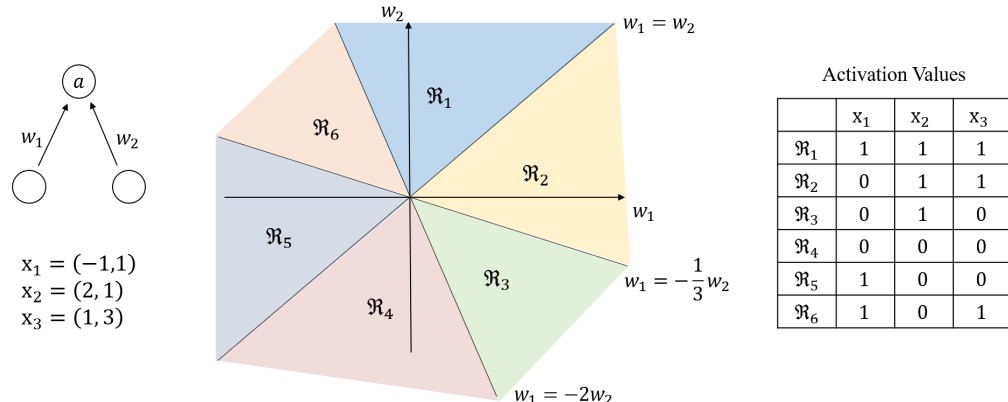

Figure 1: A simple example of the activation regions for a dataset $x_1 = [-1, 1]^T, x_2 = [2, 1]^T, x_3 = [1, 3]^T$. In each region, activation values are the same. There are six nonempty regions. The parameters on the boundaries hit the non-differentiable point of the rectified linear unit.

## 2 LOSS SURFACE DECOMPOSITION

Loss surface of deep linear networks have the following interesting properties: 1) the function is non-convex and non-concave, 2) every local minimum is a global minimum, 3) every critical point that is not a global minimum is a saddle point (Kawaguchi, 2016). This means that there is no poor local minima problem when using gradient descent methods, but such properties do not hold for nonlinear networks. We conjecture that these properties hold if activation values are fixed, and we prove it for a simple network. The loss surface of DNNs can be decomposed into regions in terms of activation values as illustrated in Figure 1.

### 2.1 MODEL AND ACTIVATION REGIONS

Let $D$ be a dataset $\{(x_1, y_1), (x_2, y_2), ..., (x_N, y_N)\}$ with $x_i \in \mathbb{R}^n$ and $y_i \in \mathbb{R}$. We define a network with one hidden layer as follows:

$$f(x_i, \theta) = v^T \sigma(W x_i + b) + c. \tag{1}$$

The model parameters are $W \in \mathbb{R}^{h \times n}$, $v \in \mathbb{R}^h$, $b \in \mathbb{R}^h$, and $c \in \mathbb{R}$ where $h$ is the number of hidden nodes. Let $\theta = [vec(W), v, b, c]^T$ collectively denote vectorized form of all the model parameters. The activation function $\sigma(x) = max(x, 0)$ is a rectified linear unit, and we abuse notation by generalizing it as an element-wise function for multidimensional inputs. Alternatively, the network can be expressed in terms of the activation values:

$$g_{\mathcal{A}}(x_i, \theta) = v^T \text{diag}(a_i)(W x_i + b) + c, \tag{2}$$

where $a_i = [a_{i1}, a_{i2}, ..., a_{ih}]^T$ is a vector of the binary activation values $a_{ij} \in \{0, 1\}$ of $i$-th data point $x_i$, and $\mathcal{A} = (a_1, a_2, ..., a_N)$ is a collection of all activation values for a given dataset $D$. We fix the activation values of the function $g_{\mathcal{A}}(x_i, \theta)$ regardless of real activation values to find out the interesting properties. The real model $f(x_i, \theta)$ agrees with $g_{\mathcal{A}}(x_i, \theta)$ only if $\mathcal{A}$ is same as the real activation values in the model.

Before we introduce a definition of the activation region, we denote $w_j^T$ as a $j$-th row of $W$, and let $\mathbb{1}(\cdot)$ be an indicator function.

**Definition 2.1** *An activation region $\mathfrak{R}_{\mathcal{A}}$ of an activation values $\mathcal{A}$ is a set of parameters satisfying $a_{ij} = \mathbb{1}(w_j^T x_i + b_j > 0)$ for all $i, j$, where $i$ is an index of data points and $j$ is an index of hidden nodes.*

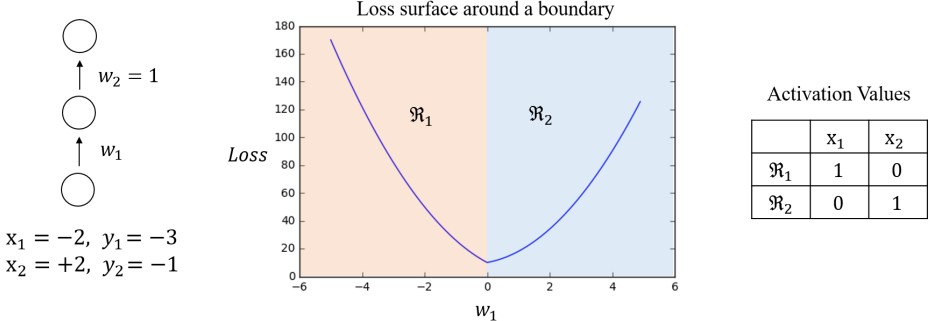

Figure 2: A simple example of a non-differentiable local minimum for a dataset $x_1 = -2, y_1 = -3, x_2 = +2, y_2 = -1$. In this example, a network is defined by $f(x) = w_2\sigma(w_1 x)$ and $w_1$ is fixed to one. The non-differentiable local minima exist in a line $w_1 = 0$ which is a boundary of the two regions. Note that if $w_1 = 0$, then $\nabla_{w_2} L_f = 0$.

We consider a general loss function called squared error loss:

$$\mathcal{L}_f(\theta) = \frac{1}{2} \sum_{i=1}^{N} (f(x_i, \theta) - y_i)^2. \tag{3}$$

The following lemma state that the local curvatures of $\mathcal{L}_{g_\mathcal{A}}(\theta)$ and $\mathcal{L}_f(\theta)$ agree in the differentiable part of $\mathfrak{R}_\mathcal{A}$.

**Lemma 2.2** *For any differentiable point $\theta \in \mathfrak{R}_\mathcal{A}$, the $\theta$ is a local minimum (saddle point) in $\mathcal{L}_f(\theta)$ if and only if it is a local minimum (saddle point) in $\mathcal{L}_{g_\mathcal{A}}(\theta)$.*

## 2.2 FUNCTION OF FIXED ACTIVATION VALUES

The function $g_\mathcal{A}(x_i, \theta)$ of fixed activation values $\mathcal{A}$ has properties similar to that of linear neural networks. If all activation values are one, then the function $g_\mathcal{A}(x_i, \theta)$ is identical to a linear neural network. In other cases, some of the parameters are inactive. The proof becomes tricky since inactive parameters are different for each data point. In case of the simple network $g_\mathcal{A}(x_i, \theta)$, we can convert it into a convex function in terms of other variables.

$$g_\mathcal{A}(x_i, \theta) = \sum_{j=1}^{h} a_{ij}(p_j^T x_i + q_j) + c, \tag{4}$$

where $p_j = v_j w_j$ and $q_j = v_j b_j$. The $v_j$ is a $j$-th scalar value of the vector $v$ and $a_{ij}$ is an activation value on a $j$-th hidden node of a $i$-th data point.

**Lemma 2.3** *The function $\mathcal{L}_{g_\mathcal{A}}(\theta)$ is a convex function in terms of $p_j$, $q_j$, and $c$.*

Note that for any $p_j$ and $q_j$, there exist $\theta$ that forms them, so the following lemma holds.

**Lemma 2.4** *The function $\mathcal{L}_{g_\mathcal{A}}(\theta)$ is minimized if and only if the gradients $\nabla_{p_j}\mathcal{L}_{g_\mathcal{A}}$, $\nabla_{q_j}\mathcal{L}_{g_\mathcal{A}}$, and $\nabla_c \mathcal{L}_{g_\mathcal{A}}$ are zeros for all $j$.*

Now we introduce the following theorem describing the important properties of the function $\mathcal{L}_{g_\mathcal{A}}(\theta)$.

**Theorem 2.5** *The function $\mathcal{L}_{g_\mathcal{A}}(\theta)$ has following properties: 1) it is non-convex and non-concave except for the case that activation values are all zeros, 2) every local minimum is a global minimum, 3) every critical point that is not a global minimum is a saddle point.*

*Sketch of proof.* A function $f(x, y) = (xy - 1)^2$ is not convex, since it has a saddle point at $x = y = 0$. Similarly, the $\mathcal{L}_{g_\mathcal{A}}(\theta)$ is a quadratic function of $v_j b_j$, so it is non-convex and non-concave. If activation values are all zeros, then $\mathcal{L}_{g_\mathcal{A}}(\theta)$ is a convex function $\frac{1}{2}\sum_{i=1}^{N}(c - y_i)^2$ with respect to $c$. If $\nabla_{\mathrm{p}_j}\mathcal{L}_{g_\mathcal{A}} = 0$ and $\nabla_{q_j}\mathcal{L}_{g_\mathcal{A}} = 0$, then $\nabla_{\mathrm{w}_j}\mathcal{L}_{g_\mathcal{A}} = 0$, $\nabla_{v_j}\mathcal{L}_{g_\mathcal{A}} = 0$ and $\nabla_{b_j}\mathcal{L}_{g_\mathcal{A}} = 0$, so the global minima are critical points. In other critical points, at least one of the gradients along $\mathrm{p}_j$ or $q_j$ is not zero. If a critical point satisfies $\nabla_{\mathrm{p}_j}\mathcal{L} \neq 0$ (or $\nabla_{q_j}\mathcal{L} \neq 0$), then it is a saddle point with respect to $\mathrm{w}_j^T$ and $v_j$ (or $b_j$ and $v_j$). The detailed proof is in the appendix.

To distinguish between the global minimum of $\mathcal{L}_{g_\mathcal{A}}(\theta)$ and $\mathcal{L}_f(\theta)$, we introduce subglobal minimum:

**Definition 2.6** *A subglobal minimum of $\mathcal{A}$ is a global minimum of $\mathcal{L}_{g_\mathcal{A}}(\theta)$.*

Some of the subglobal minima may not exist in the real loss surface $\mathcal{L}_f(\theta)$. For this kind of regions, there only exist saddle points and the parameter would move to another region by gradient descent methods without getting stuck into local minima. Since the parameter space is a disjoint union of the activation regions, the real loss surface $\mathcal{L}_f(\theta)$ is a piecewise combination of $\mathcal{L}_{g_\mathcal{A}}(\theta)$. Using Lemma 2.2 and Theorem 2.5, we conclude as follows:

**Corollary 2.7** *The function $\mathcal{L}_f(\theta)$ has following properties: 1) it is non-convex and non-concave, 2) every differentiable local minimum is a subglobal minimum, 3) every critical point that is not a subglobal minimum is a saddle point.*

We explicitly distinguish differentiable and non-differentiable local minima. The non-differentiable local minima can exist as shown in Figure 2.

## 3 EXISTENCE OF POOR LOCAL MINIMA

In this section, we answer why poor local minima do exist even in large networks. There are parameter points where all the activation values are zeros eliminating gradient flow (Swirszcz et al., 2016). This is a well-known region that forms poor and flat local minima. We introduce another kind of poor region called linear region and show that it always forms poor local minima when a dataset is nonlinear. In a more general setting, we prove that a network has every local minimum of the narrower networks of the same number of layers.

### 3.1 LINEAR REGION

There always exists a linear region where all activation values are one, and its subglobal minima stay in that region. This subglobal minimum results in an error which is same as that of linear regression, so if given dataset is nonlinear the error would be poor. We can easily spot a linear region by manipulating biases to satisfy $\mathrm{w}_j^T \mathrm{x}_i + b_j > 0$. One way of achieving this is by selecting $b_j$ as:

$$b_j = -\min_i \mathrm{w}_j^T \mathrm{x}_i + 1. \tag{5}$$

To say that the model can get stuck in the linear region, it is necessary to find the subglobal minima in that region. If $f(\mathrm{x}_i, \theta)$ is linear, then it is of form $\mathrm{u}^T \mathrm{x}_i + d$. Let $\mathrm{v}^T W = \mathrm{u}^T$, $c = -\mathrm{v}^T b + d$, and $b_j$ be same as Equation 5, then the Equation 1 is equal to $\mathrm{u}^T \mathrm{x}_i + d$ and the parameters are inside the linear region. Thus the subglobal minima do exist in the linear region. This trick is generally applicable to multilayer neural networks.

### 3.2 POOR LOCAL MINIMA ARISES FROM THE SMALLER NETWORK

Consider a multilayer neural network $f$ that has some local minima. Let $f'$ be a network constructed from $f$ by adding a single hidden node. If activation values of the new node are all zeros, then the loss values of local minima in $f'$ and $f$ are equivalent since the new parameters in $f'$ does not change the outputs of $f$. We can easily find such regions by constraining the new bias to be $b_j = -\max_i \mathrm{w}_j^T \mathrm{h}_i$ where $\mathrm{h}_i$ is a hidden layer of $i$-th data point $\mathrm{x}_i$. We numerically checked this property by identifying all subglobal minima as shown in Figure 3.

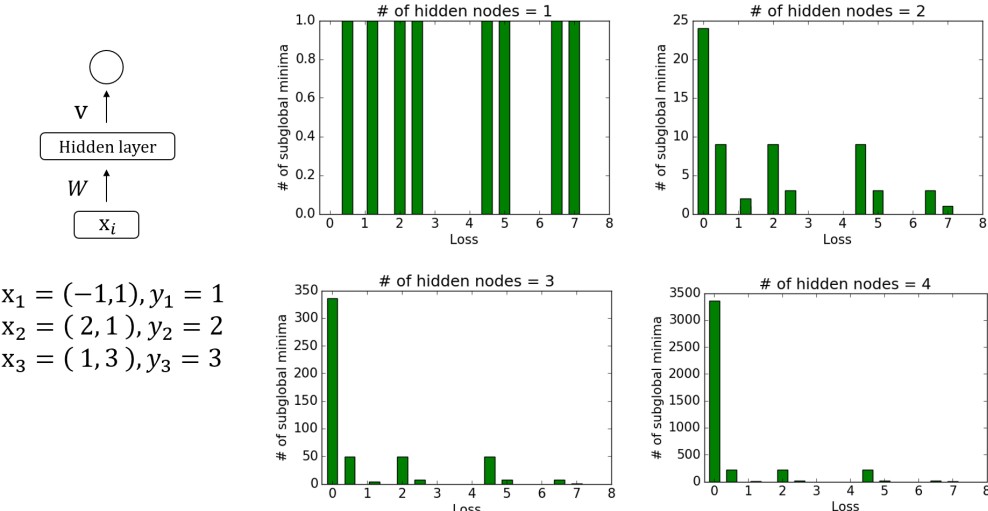

Figure 3: The histograms of subglobal minima for a dataset $x_1 = [-1,1]^T, y_1 = 1, x_2 = [2,1]^T, y_2 = 2, x_3 = [1,3]^T, y_3 = 3$. The network has one hidden layer and no biases. We increased the number of hidden nodes from one to four.

# 4 POOR REGIONS IN LARGE NETWORKS

The ratio of the poor regions decreases as the size of the network grows. We show it numerically by identifying all subglobal minima of a simple network. For the MNIST (LeCun, 1998), we estimated subglobal minima of randomly selected activation values and compared with the rich regions.

## 4.1 IDENTIFYING ALL SUBGLOBAL MINIMA

Training a neural network is known to be NP-Complete (Blum & Rivest, 1989), due to the non-convexity and infinite parameter space of DNNs. The number of possible combination of activation values has the complexity of $O(2^{Nh})$ for $f(x_i, \theta)$, so we restricted the experiments to a small size of hidden layers and datasets to find all subglobal minima in a reasonable time.

Consider the Equation 4 again. The subglobal minimum is a solution of the convex optimization for $\mathcal{L}_{g_{\mathcal{A}}}(\theta)$. To compute optimal parameters, we need to solve linear equations $\nabla_{p_j}\mathcal{L}_{g_{\mathcal{A}}} = 0$, $\nabla_{q_j}\mathcal{L}_{g_{\mathcal{A}}} = 0$, and $\nabla_c\mathcal{L}_{g_{\mathcal{A}}} = 0$. For simplicity, we assume that biases are removed, then the gradient $\nabla_{p_j}\mathcal{L}_{g_{\mathcal{A}}}$ is as follows:

$$\forall j, \nabla_{p_j}\mathcal{L}_{g_{\mathcal{A}}} = \Big(\sum_{i=1}^{N} a_{ij}x_i x_i^T\Big)p_j + \sum_{k\neq j}\Big(\sum_{i=1}^{N} a_{ij}a_{ik}x_i x_i^T\Big)p_k - \sum_{i=1}^{N} a_{ij}y_i x_i = 0. \quad (6)$$

Let $\Phi_j = \sum_{i=1}^{N} a_{ij}x_i x_i^T$, $\Lambda_{jk} = \sum_{i=1}^{N} a_{ij}a_{ik}x_i x_i^T$, and $Y_j = \sum_{i=1}^{N} a_{ij}y_i x_i$. As a result, the linear equation to solve is as follows:

$$\begin{bmatrix} \Phi_1 & \Lambda_{12} & \cdots & \Lambda_{1h} \\ \Lambda_{21} & \Phi_2 & \cdots & \Lambda_{2h} \\ \vdots & \vdots & \ddots & \vdots \\ \Lambda_{h1} & \Lambda_{h2} & \cdots & \Phi_h \end{bmatrix} \begin{bmatrix} p_1 \\ p_2 \\ \vdots \\ p_h \end{bmatrix} = \begin{bmatrix} Y_1 \\ Y_2 \\ \vdots \\ Y_h \end{bmatrix}. \quad (7)$$

Table 1: The accuracy (%) of estimated subglobal minima for MNIST.

| | $k = 16$ | $k = 32$ | $k = 64$ | $k = 128$ | $k = 256$ |
|---|---|---|---|---|---|
| Random subglobal minima | $82.59_{\pm 0.32}$ | $92.84_{\pm 0.24}$ | $98.19_{\pm 0.27}$ | $99.61_{\pm 0.07}$ | $99.60_{\pm 0.06}$ |
| Rich subglobal minima | $98.10_{\pm 0.20}$ | $99.41_{\pm 0.13}$ | $99.80_{\pm 0.06}$ | $99.88_{\pm 0.04}$ | $99.91_{\pm 0.03}$ |

The leftmost matrix in the Equation 7 is a square matrix. If it is not full rank, we compute a particular solution. Figure 3 shows four histograms of the poor subglobal minima for the different number of hidden nodes. As shown in the histograms, gradient descent based methods are more likely to avoid poor subglobal minima in larger networks. It also shows that the subglobal minima arise from the smaller networks. Intuitively speaking, adding a node provides a downhill path to the previous poor subglobal minima without hurting the rich subglobal minima in most cases.

## 4.2 EXPERIMENTS ON MNIST

For more realistic networks and datasets, we conducted experiments on MNIST. We used networks of two hidden layers consisting of $2k$ and $k$ nodes respectively. The networks use biases, softmax outputs, cross entropy loss, mini-batch size of 100, and Adam Optimizer (Kingma & Ba, 2015). Assuming that the Corollary 2.7 holds for multilayer networks, the subglobal minima can be estimated by gradient descent methods. It is impossible to compute all of them, so we randomly selected various combinations of activation values with $P(a = 1) = P(a = 0) = 0.5$. Then we removed rectified linear units and multiplied the fixed activation values as follows:

$$h_{\mathcal{A}}(\mathrm{x}_i, \theta) = \mathrm{diag}(\mathrm{a}_{i2})(W_2 \mathrm{diag}(\mathrm{a}_{i1})(W_1 \mathrm{x}_i + \mathrm{b}_1) + \mathrm{b}_2), \tag{8}$$

where $h_{\mathcal{A}}$ is the output of the second hidden layer. The rich subglobal minima were estimated by optimizing the real networks since it would end up in one of the subglobal minima that exist in the real loss surface. The experiments were repeated for 100 times, and then we computed mean and standard deviation. The results are shown in Table 1 and it implies that most of the regions in the large networks are rich, whereas the small networks have few rich regions. In other words, it is more likely to end up in a rich subglobal minimum in larger networks.

## 5 RELATED WORKS

(Baldi & Hornik, 1989) proved that linear networks with one hidden layer have the properties of the Theorem 2.5 under minimal assumptions. Recently, (Kawaguchi, 2016) proved that it also holds for deep linear networks. Assuming that the activation values are drawn from independent Bernoulli distribution, a DNN can be mapped to a spin-glass Ising model in which the number of local minima far from the global minima diminishes exponentially with the size of the network (Choromanska et al., 2015). Under same assumptions in (Choromanska et al., 2015), the effect of nonlinear activation values disappears by taking expectation, so nonlinear networks satisfy the same properties of linear networks (Kawaguchi, 2016).

Nonlinear DNNs usually do not encounter any significant obstacles on a single smooth slope path (Goodfellow et al., 2014), and (Dauphin et al., 2014) explained that the training error at local minima seems to be similar to the error at the global minimum which can be understood via random matrix theory. The volume of differentiable sub-optimal local minima is exponentially vanishing in comparison with the same volume of global minima under infinite data points (Soudry & Hoffer, 2017). Although a number of specific example of local minima can be found in DNNs (Swirszcz et al., 2016), it seems plausible to state that most of the local minima are near optimal.

As the network width increases, we are more likely to meet a random starting point from which there is a continuous, strictly monotonically decreasing path to a global minimum (Safran & Shamir, 2016). Similarly, the starting point of the DNNs approximate a rich family of hypotheses precisely (Daniely et al., 2016). Another explanation is that the level sets of the loss become connected as the network is increasingly overparameterized (Freeman & Bruna, 2015). These works are analogous to our results

showing that the parameters would end up in one of the subglobal minima which are similar to the global minima.

## 6 DISCUSSION AND CONCLUSION

We conjecture that the loss surface is a disjoint union of activation regions where every local minimum is a subglobal minimum. Using the concept of loss surface decomposition, we studied the existence of poor local minima and experimentally investigated losses of subglobal minima. However, the structure of non-differentiable local minima is not yet well understood yet. These non-differentiable points exist within the boundaries of the activation regions which can be obstacles when using gradient descent methods. Further work is needed to extend knowledge about the local minima, activation regions, their boundaries.

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

## APPENDIX A    PROOFS OF LEMMAS AND THEOREM

### A.1    PROOF OF LEMMA 2.2

Let $\theta \in \mathfrak{R}_{\mathcal{A}}$ be a differentiable point, so it is not in the boundaries of the activation regions. This implies that $w_j^T x_i + b_j \neq 0$ for all parameters. Without loss of generality, we assume $w_j^T x_i + b_j < 0$. Then there exist $\epsilon > 0$ such that $w_j^T x_i + b_j + \epsilon < 0$. This implies that small changes in the parameters for any direction does not change the activation region. Since $\mathcal{L}_f(\theta)$ and $\mathcal{L}_{g_{\mathcal{A}}}(\theta)$ are equivalent in the region $\mathfrak{R}_{\mathcal{A}}$, the local curvatures of these two function around the $\theta$ are also the same. Thus, the $\theta$ is a local minimum (saddle point) in $\mathcal{L}_f(\theta)$ if and only if it is a local minimum (saddle point) in $\mathcal{L}_{g_{\mathcal{A}}}(\theta)$.

### A.2    PROOF OF LEMMA 2.3

Clearly, $(g_{\mathcal{A}}(x_i, \theta) - y_i)^2$ is convex in terms of $g_{\mathcal{A}}(x_i, \theta)$. Since $g_{\mathcal{A}}(x_i, \theta)$ is a linear transformation of $p_j$, $q_j$, and $c$, the $(g_{\mathcal{A}}(x_i, \theta) - y_i)^2$ is convex in terms of $p_j$, $q_j$, and $c$. Summation of convex functions is convex, so the lemma holds.

### A.3    PROOF OF THEOREM 2.5

**(1)** Assume that activation values are not all zeros, and then consider the following Hessian matrix evaluated from $v_j$ and $b_j$ for some non-zero activation values $a_{ij} > 0$:

$$
\begin{aligned}
\nabla^2_{v_j, b_j} \mathcal{L}_{g_{\mathcal{A}}}(\theta) &= \begin{bmatrix} \frac{\partial^2 \mathcal{L}_{g_{\mathcal{A}}}}{\partial v_j^2} & \frac{\partial^2 \mathcal{L}_{g_{\mathcal{A}}}}{\partial v_j \partial b_j} \\ \frac{\partial^2 \mathcal{L}_{g_{\mathcal{A}}}}{\partial v_j \partial b_j} & \frac{\partial^2 \mathcal{L}_{g_{\mathcal{A}}}}{\partial b_j^2} \end{bmatrix} \\
\frac{\partial^2 \mathcal{L}_{g_{\mathcal{A}}}}{\partial v_j^2} &= \sum_{i=1}^{N} a_{ij} (w_j^T x_i + b_j)^2 \\
\frac{\partial^2 \mathcal{L}_{g_{\mathcal{A}}}}{\partial b_j^2} &= \sum_{i=1}^{N} a_{ij} v_j^2 \\
\frac{\partial^2 \mathcal{L}_{g_{\mathcal{A}}}}{\partial v_j \partial b_j} &= \sum_{i=1}^{N} a_{ij} (v_j (w_j^T x_i + b_j) + g_{\mathcal{A}}(x_i, \theta) - y_i)
\end{aligned}
\tag{9}
$$

Let $v_j = 0$ and $b_j = 0$, then two eigenvalues of the Hessian matrix are as follows:

$$
2\lambda = \frac{\partial^2 \mathcal{L}_{g_{\mathcal{A}}}}{\partial v_j^2} \pm \sqrt{\left(\frac{\partial^2 \mathcal{L}_{g_{\mathcal{A}}}}{\partial v_j^2}\right)^2 + 4\left(\frac{\partial^2 \mathcal{L}_{g_{\mathcal{A}}}}{\partial v_j \partial b_j}\right)^2}.
\tag{10}
$$

There exist $c > 0$ such that $g_{\mathcal{A}}(x_i, \theta) > y_i$ for all $i$. If we choose such $c$, then $\frac{\partial^2 \mathcal{L}_{g_{\mathcal{A}}}}{\partial v_j \partial b_j} > 0$ which implies that two eigenvalues are positive and negative. Since the Hessian matrix is not positive semidefinite nor negative semidefinite, the function $\mathcal{L}_{g_{\mathcal{A}}}(\theta)$ is non-convex and non-concave.

**(2, 3)** We organize some of the gradients as follows:

$$\nabla_{b_j} \mathcal{L}_{g_{\mathcal{A}}}(\theta) = v_j \sum_{i=1}^{N} (g_{\mathcal{A}}(\mathrm{x}_i, \theta) - y_i) a_{ij}$$

$$\nabla_{\mathrm{w}_j} \mathcal{L}_{g_{\mathcal{A}}}(\theta) = v_j \sum_{i=1}^{N} (g_{\mathcal{A}}(\mathrm{x}_i, \theta) - y_i) a_{ij} \mathrm{x}_i$$

$$\nabla_{v_j} \mathcal{L}_{g_{\mathcal{A}}}(\theta) = \sum_{i=1}^{N} (g_{\mathcal{A}}(\mathrm{x}_i, \theta) - y_i) a_{ij} (\mathrm{w}_j^T \mathrm{x}_i + b_j) \tag{11}$$

$$\nabla_{\mathrm{p}_j} \mathcal{L}_{g_{\mathcal{A}}}(\theta) = \sum_{i=1}^{N} (g_{\mathcal{A}}(\mathrm{x}_i, \theta) - y_i) a_{ij} \mathrm{x}_i$$

$$\nabla_{q_j} \mathcal{L}_{g_{\mathcal{A}}}(\theta) = \sum_{i=1}^{N} (g_{\mathcal{A}}(\mathrm{x}_i, \theta) - y_i) a_{ij}.$$

We select a critical point $\theta^*$ where $\nabla_{\mathrm{w}_j} \mathcal{L}_{g_{\mathcal{A}}}(\theta^*) = 0$, $\nabla_{v_j} \mathcal{L}_{g_{\mathcal{A}}}(\theta^*) = 0$, $\nabla_{b_j} \mathcal{L}_{g_{\mathcal{A}}}(\theta^*) = 0$, and $\nabla_c \mathcal{L}_{g_{\mathcal{A}}}(\theta^*) = 0$ for all $j$.

**Case 1)** Assume that $\nabla_{\mathrm{p}_j} \mathcal{L}_{g_{\mathcal{A}}}(\theta^*) = 0$ and $\nabla_{q_j} \mathcal{L}_{g_{\mathcal{A}}}(\theta^*) = 0$ for all $j$. These points are global minima, since $\nabla_c \mathcal{L}_{g_{\mathcal{A}}}(\theta^*) = 0$ and $\mathcal{L}_{g_{\mathcal{A}}}(\theta)$ is convex in terms of $\mathrm{p}_j$, $q_j$, and $c$.

**Case 2)** Assume that there exist $j$ such that $\nabla_{\mathrm{p}_j} \mathcal{L}_{g_{\mathcal{A}}}(\theta^*) \neq 0$. Since $\nabla_{\mathrm{w}_j} \mathcal{L}_{g_{\mathcal{A}}}(\theta^*) = v_j \nabla_{\mathrm{p}_j} \mathcal{L}_{g_{\mathcal{A}}}(\theta^*) = 0$, the $v_j$ is zero. If $v_j = 0$, then:

$$\nabla_{\mathrm{w}_j}^2 \mathcal{L}_{g_{\mathcal{A}}}(\theta^*) = 0$$

$$\nabla_{v_j} \nabla_{\mathrm{w}_j} \mathcal{L}_{g_{\mathcal{A}}}(\theta^*) = \sum_{i=1}^{N} ((g_{\mathcal{A}}(\mathrm{x}_i, \theta) - y_i) + v_j(\mathrm{w}_j^T \mathrm{x}_i + b_j)) a_{ij} \mathrm{x}_i$$

$$= \sum_{i=1}^{N} (g_{\mathcal{A}}(\mathrm{x}_i, \theta) - y_i) a_{ij} \mathrm{x}_i$$

$$= \nabla_{\mathrm{p}_j} \mathcal{L}_{g_{\mathcal{A}}}(\theta^*) \tag{12}$$

$$\neq 0$$

$$\nabla_{v_j}^2 \mathcal{L}_{g_{\mathcal{A}}}(\theta^*) = \sum_{i=1}^{N} (\mathrm{w}_j^T \mathrm{x}_i + b_j)^2 a_{ij}$$

$$\geq 0.$$

There exist an element $w^*$ in $\mathrm{w}_j$ such that $\nabla_{v_j} \nabla_{w^*} \mathcal{L}_{g_{\mathcal{A}}}(\theta^*) \neq 0$. Consider a Hessian matrix evaluated from $w^*$ and $v_j$. Analogous to the proof of (1), this matrix is not positive semidefinite nor negative semidefinite. Thus $\theta^*$ is a saddle point.

**Case 3)** Assume that there exist $j$ such that $\nabla_{q_j} \mathcal{L}_{g_{\mathcal{A}}}(\theta^*) \neq 0$. Since $\nabla_{b_j} \mathcal{L}_{g_{\mathcal{A}}}(\theta^*) = v_j \nabla_{q_j} \mathcal{L}_{g_{\mathcal{A}}}(\theta^*) = 0$, the $v_j$ is zero. Analogous to the Case 2, a Hessian matrix evaluated from $b_j$ and $v_j$ is not positive semidefinite nor negative semidefinite. Thus $\theta^*$ is a saddle point.

As a result, every critical point is a global minimum or a saddle point. Since $\mathcal{L}_{g_{\mathcal{A}}}(\theta)$ is a differentiable function, every local minimum is a critical point. Thus every local minimum is a global minimum.

