# OpenReview forum: "Understanding Local Minima in Neural Networks by Loss Surface Decomposition"
_ICLR.cc/2018/Conference — Reject_

### Official Review · AnonReviewer2 · 2017-11-27

**Rating:** 4
**Confidence:** 4

**Review:**

This paper proposes to study the loss surfaces of neural networks with ReLU activations by viewing the loss surface as a sum of piecewise linear functions at each point in parameter space, i.e. one piecewise linear function per sample. The main result is that every local minimum of the total surface is a global minimum of the region where the ReLU activations corresponding to each sample do not change.

Quality:
- The paper's claims are correct, however most of the theoretical results follow easily from the definitions and I don't see how they are very useful. What is interesting about other recent theoretical works, which show (subject to various assumptions) that local minima are roughly equivalent to the global minimum, is that they compare local minima across all regions of the parameter space and show they are similar. Here, the results only hold within each local region of the space, and they don't say anything about how the global minima in different regions compare in terms of loss. Knowing that a local minumum is a global minimum of a local region is not very useful because the global minimum of that local region could still be much worse than that of the other regions.


Clarity:
- The main claims/results in the paper are not stated very clearly, and the authors are not clear about what the contributions of the paper are or why they are useful.

Originality:
- Studying loss surfaces by viewing ReLU networks as piecewise linear functions is by now standard.

Significance:
- It is not clear how these results may be applied in practice or open new directions for future theoretical work.

---

### Official Review · AnonReviewer3 · 2017-11-27
**Simple but interesting**

**Rating:** 5
**Confidence:** 4

**Review:**

The authors propose investigating regions of the the parameter space under which the activations (over the entire training data set) remain unchanged.  They conjecture, and then attempt to argue for a simple network, that, over these regions, the loss function exhibits nice properties:  all local minima are global minima, all other local optima are saddle points, and the function is neither convex nor concave on these regions.  The proof of this statement seems relatively straightforward and appears to be correct.  Unfortunately it only applies to a special case.  Second, the authors argue that the loss function for their simple network has poor local minima.  Finally, the authors conclude with a simple set of experiments exploring the accuracy of random activations.  Overall, I found the main idea of the paper relatively straightforward, but the presentation is a bit awkward in places.

I think the work is heading in an interesting direction, but I found it somewhat incremental.  It's nice to know that the loss function (squared loss in this case) has these properties, but as there are exponentially many regions corresponding to the different activations, it is unclear what the practical consequences of these theoretical observations are.  Could the authors elaborate on this?

Another question:  is it really true that the non-differentiability of the functions involved creates significant issues in practice  (not theoretically) - isn't the set of all points with this property of measure zero?

---

### Official Review · AnonReviewer1 · 2017-11-28
**While addressing an interesting problem, the theoretical results in this paper are not of sufficient quality or relevance**

**Rating:** 4
**Confidence:** 4

**Review:**

This paper attempts to extend analytical results pertaining to the loss surface of linear networks to a nonlinear network with a single hidden ReLU layer.  Unfortunately though, at this point I feel that the theoretical results, which constitute the majority of the paper, are of limited novelty and/or significance.  However, I still remain very open to counterarguments to this opinion and the points raised below.

First, I don't believe that Lemma 2.2 is precisely true, at least as currently stated.  In particular, it would appear that L_f could have a differentiable local minima that is only a saddle point in L_gA.  For example, if there is a differentiable valley in L_f that terminates on the boundary of an activation region, then this phenomena could occur, since a local-minima-creating boundary in L_f might just lead to a saddle point in L_gA.  Regardless, the basic premise of this result is quite straightforward anyway.

Turning to Lemma 2.3 and 2.4, I don't understand the relevance of these results.  Where are they needed later or applied?  Additionally, Theorem 2.5 is very related to results already proven for linear networks in earlier work (Kawaguchi, 2016), so there is little novelty here.

There also seem to be issues with Corollary 2.7, which as an aggregation result can be viewed as the main contribution of the paper.  Part (1) of this corollary is obvious.  Part (2) depends on Lemma 2.2, which as stated previously may be problematic.  Most seriously though, Part (3) only considers critical points (i.e., derivative equal to zero), not local minima occurring at non-differentiable locations.  To me this greatly mutes the value of this result, and the contribution of the paper overall, because local minimum are *very* likely to occur on the boundary between activation regions at non-differentiable points (e.g. as in Figure 2).  I therefore don't understand the utility of only considering the differentiable local minima.

Overall though, the main point that within areas of fixed activation the network behaves much like a linear network (with all local minima also global minima when constrained within each region), is not especially noteworthy, because it provides no pathway for comparing minima from different activation regions, which is the main problem to begin with.

Beyond this, the paper makes a few less-technical observations regarding bad local minima.  For example, in Section 3.1 the argument is made that the linear region created when all activations are equal to one, will have a local minimum, and this minimum might be suboptimal.  However, these arguments pertain to the surrogate function L_gA, and if the minima to L_gA occurs on the boundary to another activation region, then this solution might not be a local minima to L_f, the real objective we care about.  Am I missing something here?

As for Section 4.2, the paper needs to do a better job of explaining exactly what is been shown in Table 2.  I can maybe guess, but it is not at all clear what the accuracy percentage is referring to, nor precisely how rich and random minima are computed.  Also, the assumption that P(a = 1) = P(a = 0) = 0.5 is not very realistic, although admittedly this type of simplification is sometimes adopted in the literature.

Minor comment:
* Near the beginning in the introduction, it is claimed that "the vanishing gradient problem has been solved by using rectified linear units."  This is not actually true, and portends problematic claims later in the paper.

---

### Decision · Program_Chairs · 2018-01-29
**ICLR 2018 Conference Acceptance Decision**

**Decision:**

Reject

**Comment:**

The reviewers are unanimous in their opinion that the theoretical results in this paper are of limited novelty and significance. Several parts of the paper are not presented clearly enough. As such the paper is not ready for ICLR-2018 acceptance.